# Tunisian Multicenter Study on the Prevalence of Colistin Resistance in Clinical Isolates of Gram Negative Bacilli: Emergence of *Escherichia coli* Harbouring the *mcr-1* Gene

**DOI:** 10.3390/antibiotics11101390

**Published:** 2022-10-11

**Authors:** Sana Ferjani, Elaa Maamar, Asma Ferjani, Khaoula Meftah, Hager Battikh, Besma Mnif, Manel Hamdoun, Yosra Chebbi, Lamia Kanzari, Wafa Achour, Olfa Bahri, Adenene Hammami, Meriam Zribi, Hanen Smaoui, Ilhem Boutiba-Ben Boubaker

**Affiliations:** 1Faculty of Medicine of Tunis, University of Tunis El Manar, LR99ES09, Tunis Rue Djebal Lakhdar 1006, Tunisia; 2Charles Nicolle Hospital, Laboratory of Microbiology, Boulevard 9 April, Tunis 1006, Tunisia; 3Laboratory of Microbiology, Children’s Hospital of Tunis, Boulevard 9 April, Tunis 1006, Tunisia; 4Microbiology Laboratory, Rabta University Hospital, Rue Jabbari, Tunis 1007, Tunisia; 5Laboratory of Microbiology, Habib Bourguiba University Hospital, Route de l’Ain, Sfax 3000, Tunisia; 6Research Laboratory for Microorganisms and Human Disease, University of Sfax, Avenue Majida Boulila, Sfax 3029, Tunisia; 7Aziza Othmana Hospital, Laboratoire de Microbiologie-Biochimie, Bab Menara Tunis 1008, Tunisia; 8Faculty of Medicine of Tunis, University of Tunis El Manar, LR16SP01, Tunis Rue Djebal Lakhdar 1006, Tunisia; 9National Bone Marrow Transplant Center, Laboratory Ward, Tunis Rue Djebal Lakhdar 1006, Tunisia; 10Faculty of Medicine of Tunis, Tunis El Manar University, LR18ES39, Tunis Rue Djebal Lakhdar 1006, Tunisia

**Keywords:** colistin resistance, Gram negative bacteria, *mcr-1* gene, multi-drug resistant bacteria

## Abstract

Background: Actually, no data on the prevalence of plasmid colistin resistance in Tunisia are available among clinical bacteria. Objectives: This study aimed to investigate the current epidemiology of colistin resistance and the spread of the *mcr* gene in clinical Gram-negative bacteria (GNB) isolated from six Tunisian university hospitals. Methods: A total of 836 GNB strains were inoculated on COL-R agar plates with selective screening agar for the isolation of GNB resistant to colistin. For the selected isolates, *mcr* genes, beta-lactamases associated-resistance genes and molecular characterisation were screened by PCRs and sequencing. Results: Colistin-resistance was detected in 5.02% (42/836) of the isolates and colistin-resistant isolates harboured an ESBL (*bla*_CTX-M-15_) and/or a carbapenemase (*bla*_OXA-48_, *bla*_VIM_) encoding gene in 45.2% of the cases. The *mcr*-1 gene was detected in four *E. coli* isolates (0.59%) causing urinary tract infections and all these isolates also contained the *bla*_TEM-1_ gene. The *bla*_CTX-M-15_ gene was detected in three isolates that also carried the IncY and IncFIB replicons. The genetic environment surrounding the *mcr*-carrying plasmid indicated the presence of *pap-2* gene upstream *mcr*-1 resistance marker with unusual missing of ISApl1 insertion sequence. The Conclusions: This study reports the first description of the *mcr*-1 gene among clinical *E. coli* isolates in Tunisia and provides an incentive to conduct routine colistin susceptibility testing in GNB clinical isolates.

## 1. Introduction

Colistin has been considered one of the last-resort antibiotics, for the treatment of serious infections caused by extensively drug resistant Gram negative bacteria (GNB) mainly carbapenem-resistant *Enterobacteriaceae* (CRE). In recent years, multiple studies have described the rapid increase in the prevalence of colistin resistance among *Enterobacteriaceae*, *Acinetobacter* spp. and *Pseudomonas* spp. [1,2]. In 2015, Liu et al. identified the mobile colistin resistance gene, *mcr*-1, primarily in *Enterobacteriaceae*, mainly *Escherichia coli* and *K. pneumoniae* [3]. Since then, several reports have indicated that *mcr*-1 has silently spread worldwide since 1980. To date, nine different plasmid-encoded colistin resistance genes have been described (*mcr*-1 through *mcr*-9, that have been isolated from bacteria in human, animal and environmental samples [4]. Of particular concern is the dissemination of the *mcr* gene in CRE and/or extended-spectrum-beta-lactamase (ESBL)-producing GNB, potentially leading to pan-drug-resistant isolates [3,5]. Until the discovery of *mcr*-1, all reported polymyxin resistance mechanisms were chromosomally mediated, due to mutation and regulatory changes [3], and had never been reported to be transmitted horizontally [3].

With the emergence of *mcr* determinants in bacteria from animals and humans, as well as the continuous and global use of colistin in both clinical and non-clinical settings, the surveillance of *mcr* variants in bacteria from clinical, veterinary and environmental sources is needed. Recently, there have been numerous studies on the emergence of the *mcr*-1 gene in humans, animals, food and environment worldwide [2,3,4,5]. In Tunisia, several studies confirmed the presence of this gene in bacteria isolated from healthy and sick animals [6,7,8,9], but to the best of our knowledge, there has not been a report of infection caused by a pathogen harbouring an *mcr* gene. [1,10,11]. Besides, colistin susceptibility testing was routinely performed only for carbapenem resistant GNB, which may contribute to the unawareness of resistance.

The aim of this work was to investigate the prevalence of colistin-resistance and to determine the occurrence of the *mcr* genes in clinical GNB isolates collected from human samples in six Tunisian university hospitals.

## 2. Materials and Methods

### 2.1. Study Design 

During a two-month period (March–April 2019), a multicenter-prospective study including all GNB strains was conducted in six different university hospitals. Five were located in the north of Tunisia [Bechir Hamza Children’s Hospital of Tunis (CH) (hospital 1; 335 beds), Charles Nicolle Hospital of Tunis (CNH) (hospital 2; 1094 beds), the Rabta Hospital (RH) (hospital 3; 930 beds), Aziza Othmana Hospital of Tunis (AOH) (hospital 4; 136 beds), the National Bone Marrow Transplant Center (NBMTC) (hospital 5; 56 beds) and one in the south of the country (Habib Bourguiba University Hospital of Sfax (HBS) (hospital 6; 540 beds)] (Table 1).

### 2.2. Sample Processing and Microbial Study

A total of 836 GNB strains, consecutively isolated from clinical samples, were collected from several samples (mainly from urine, blood, sputum and pus). All specimens were treated according to standard microbiological procedures (REMIC) [12]. Only one isolate per patient was included and isolates from colonization screening studies were not considered. Species intrinsically resistant to colistin were excluded. Isolates were firstly identified based on colony morphology and Gram stain. Then all colonies were tested for oxidase by adding bacterial inoculum on the cotton tipped swab already impregnated by the N,N,N,N tetramethyl-paraphenylenediamine reagent. According to the results of the Gram stain and oxidase test, the API 20E and API NE systems (bioMérieux, Marcy l’Etoile, France) were used for biochemical identification of *Enterobacteriaceae* and non-*Enterobacteriaceae*, respectively. API strips test containing 20 wells with dehydrated substrates to detect enzymatic activity. A bacterial suspension was used to rehydrate strip wells before incubation. All results from the tests were compiled to obtain a profile number, which is then compared with profile numbers in the API-Web software to determine the identification of the bacterial species. *E. coli* ATCC 25922 and *P. aeruginosa* ATCC27853 were used as control strains. 

All participants followed the same protocol for the initial colistin resistance (COL-R) GNB screening. A bacterial suspension of 0.5 McFarland was prepared from isolated colonies on trypticase soy agar. Then a loopful (10 µL) of each GNB suspension was streaked onto a selective agar medium CHROMagar™ COL-APSE (CHROMagar, Paris, France) and incubated overnight at 37 °C. The results were defined by observing different coloured colonies, using the manufacturer’s interpretation criteria for identifying COL-R GNB. To avoid the inoculum effect problem, only colonies that have grown in the second dial were taken. 

All the selected isolates were stored in brain heart infusion broth with 20% glycerol at −80 °C during the data collection period. Then, they were sent to the National Reference Laboratory at CNH for further investigations (colistin susceptibility testing, *mcr*-screening and genotyping, etc).

#### 2.2.1. Antimicrobial Susceptibility Testing

Antimicrobial susceptibility tests were performed for all isolates using the disk diffusion method on Mueller-Hinton (MH) agar plates (Bio-Rad, Marnes-la-Coquette, France) according to the CA-SFM guidelines (http://www.sfm-microbiologie.org/, accessed on 1 June 2019).

Minimum Inhibitory Concentrations (MICs) of colistin were determined, for all the screened GNB isolates, by the standard broth microdilution (BMD) method according to the Clinical and Laboratory Standards Institute (CLSI) guidelines [M7-A10] [13]. Susceptibility to colistin sulfate (Sigma-Aldrich, St. Louis, MO, USA) was tested over a range of two-fold dilutions (0.12–64 mg/L). All experiments were repeated in triplicate. *E. coli* ATCC 25922, *P. aeruginosa* ATCC 27853 and *E. coli* C6944 (harbouring *mcr*-1 gene, colistin MIC = 8 (11)) were used as quality control strains.

#### 2.2.2. Screening for mcr Genes and Determination of Its Genetic Environment

A multiplex PCR assay was used to detect the mobile colistin resistance genes in all colistin resistant isolates, *mcr*-1 to *mcr*-9, using already described protocols and primers (Appendix A) [4,14]. The genetic environment of the *mcr*-1 gene was determined by PCR mapping using specific primers as previously described (Appendix A) [6,15]. Indeed, To screen for the upstream presence of *ISApl1*, *mcr-1*-positive strains were examined with PCR using primers ISApl1-mcr-F and ISApl1-mcr-R.

#### 2.2.3. Detection and Characterization of Beta-Lactamase Genes

Colistin resistant isolates showing beta-lactams resistance phenotype were screened by multiplex PCR assays for genes encoding the most frequent beta-lactamase: OXA-1-like broad-spectrum beta-lactamases, extended-spectrum beta-lactamases (*bla*_CTX-M_, *bla*_TEM_, and *bla*_SHV_), plasmid-mediated AmpC beta-lactamases (*bla*_MOX_, *bla*_CIT_, *bla*_DHA_, *bla*_ACC_, *bla*_EBC_ and *bla*_FOX_) and class A, B and D carbapenemases (*bla*_GES_, *bla*_KPC_, *bla*_NDM_, *bla*_VIM_, *bla*_IMP_ and *bla*_OXA-48_, genes) (Appendix A) [16,17].

#### 2.2.4. Characterization of Isolates Harbouring mcr Gene

##### Conjugation Assays and Plasmid Replicon Type

The transfer of the *mcr* gene was investigated by conjugation assays using *E. coli* J53 as the recipient strain. MH agar plates (Bio-Rad) containing sodium azide (200 mg/L) supplemented with colistin sulfate (2 mg/L) and/or cefotaxime (2 mg/L) was used to isolate transconjugants. PCR-based replicon typing method (PBRT) was used to determine the plasmid incompatibility groups (Inc), among donor and transconjugants strains (Appendix A) [18]. 

##### Integrons Detection

*int*1, *int*2 and *int*3 genes encoding integrases of class 1, class 2 and class 3 integrons, respectively were investigated by PCR (Appendix A) [19]. 

##### Detection of *E. coli* Phylogenetic Groups 

Phylogenetic groups were determined by multiplex PCR targeting four genes (*chu*A, *yja*A, TspE4.C2, and *arp*A) allowing the isolates to be classified into seven main phylogenetic groups (A, B1, B2, C, D, E, and F) and cryptic clade I as described previously (Appendix A) [20].

### 2.3. Ethical Approval

The strains investigated in this study were issued from bacteriological diagnostic of pathogenic specimens routinely received by the different microbiology departments participating in this study. Therefore ethical approval and consent were not required.

## 3. Results

Over the study period, 836 GNB isolates were collected, among them, 77 (9.2%) showed growth on CHROMagar™ COL-APSE medium and 42 [5.02% (90% CI 3.8–6.3)] were colistin resistant according to MICs results. Colistin resistant strains are distributed as shown in Table 2. 

Colistin resistance was detected in 28 isolates of *P. aeruginosa* [9.3% (90% CI 6.5–12.1)], in 7 *E. coli* [3% (90% CI 1.2–4.8)] and 5 *K. pneumoniae* [2.7% (90% CI 0.8–4.7)]. Resistance was also detected in *E. cloacae* (*n* = 1) and *Raoutella terrigena* (*n* = 1) (Table 2). The range of colistin MICs in all isolates was 4–128 mg/L. Resistant isolates were mainly recovered from wounds (33.3%) and urine (28.6%), followed by sputum (11.9%) and blood (7.1%).

Among the 28 colistin-resistant *P. aeruginosa*, 10 (35.7%) were carbapenemase producers harbouring the *bla*_VIM_ gene. Three (42.8%) of the colistin-resistant *E. coli* isolates were extended spectrum β-lactamase (ESBL)-producers harbouring the *bla*_CTX-M-15_ gene. *K. pneumoniae* (*n* = 5) and *R. terrigena* (*n* = 1) isolates carried both an ESBL and a carbapenemase encoded, respectively, by *bla*_CTX-M-15_ and *bla*_OXA-48_.

The *mcr*-1 gene was detected in four *E. coli* isolates with colistin MIC ranging from 16 mg/L to 32 mg/L. They were responsible for urinary tract infections, community-acquired in three of four cases (Table 3). 

All *mcr*-1 positive isolates (*n* = 4) co-harboured *bla*_TEM-1_, while three of them (E2, E3 and E4) were ESBL producers co-harbouring the *bla*_CTX-M-15_ gene.

The conjugation assay was positive for only one (E1) of the four *E. coli* isolates carrying *mcr*-1 despite repeated attempts. The presence of the *mcr-1* and *bla*_TEM-1_ genes was confirmed in the *E. coli* J53 transconjugant by standard PCR. Other antibiotic-resistant phenotypes (fluoroquinolones and tetracyclin) could not be co-transferred with colistin. Colistin MIC of the transconjugant showed a 4-fold decrease in the MICs in comparison with the donor *E. coli* « E1 » (16 mg/L). The analysis of the transconjugant revealed that the *mcr*-1 plasmid was untypeable by PBRT. The genetic environment surrounding this plasmid was *mcr*-1-*pap*2. PCR results showed missing ISApl1 in the upstream region of *mcr*-1.

The three other *mcr*-1 positive *E. coli* isolates (E2, E3 and E4) carried IncY and IncFIB replicons. Their phylogenetic analysis revealed that they belonged to the phylogroup D. The remaining *mcr*-1-carrying *E. coli* was untypeable.

## 4. Discussion

The present study described the prevalence of colistin resistance in clinical GNB isolates from six different university hospitals in Tunisia as well as the emergence of the *mcr* gene. Colistin resistance was found in 5.02% of the isolates. Colistin susceptibility is typically tested in multidrug resistant clinical isolates for therapeutic reasons. Thus, until now the prevalence of colistin resistance among clinical isolates still remains underestimated. In our study, only 45.2% of colistin-resistant isolates harboured ESBL and/or carbapenemase genes.

This study reported, for the first time, the emergence of human clinical *E. coli* isolates harbouring the *mcr-1* gene causing urinary tract infections in Tunisia with a high prevalence of 10% (4/42) among colistin-resistant GNB compared with prevalence rates reported from other studies [3,21]. To date, in our country, multiple studies have reported the dissemination of colistin resistance among GNB, mainly among *K. pneumoniae* causing nosocomial infections [1,11], which was mostly linked to alterations identified within the *mgrB* gene but not related to plasmid-encoded colistin resistance genes *mcr*-*1* to *mcr*-*9* [1,10,11].

Previous Tunisian studies have described the dissemination and the high prevalence of *E. coli* strains harbouring *mcr-1* in animals, including healthy food-producing animals and diseased animals [6,7,8]. Overall, with the extensive use of colistin in hospitals, colistin resistance has already spread worldwide [2]. Besides, there are also reports of colistin resistance in humans who have not previously received this drug. Colistin use in animals suggests that animals may be an important source of transmission of colistin resistance to humans [2]. 

According to the medical record review about patients who acquired *E. coli* harbouring the *mcr-1* gene, the onset of patient symptoms occurred before the consultation of emergency and urology wards. These data suggested the community origin of the urinary tract infection caused by isolates harbouring the *mcr-1* gene. In order to support this hypothesis concluded during this short surveillance period, further studies are needed to evaluate the spread of plasmid colistin resistance outside hospitals and control their transmission. 

The four *mcr-1* positive *E. coli* isolates were MDR and carried the *bla*_TEM-1_ gene, while three of them were ESBL positive, co-harbouring the *bla*_CTXM-15_ gene. The co-existence of *mcr*-1 with ESBL has been frequently reported [9,21]. One study suggests that the co-occurrence of *mcr*-1 and *bla*_CTXM-15_ is perhaps due to intricate genetic actions taken under antibiotic pressure due to the integration of *mcr*-1 into the bacterial chromosome in some strains. Most of the *mcr*-1 positive strains have been proven to be carrying different beta-lactamase genes including pAmpC, ESBL, and carbapenemase genes [6,9], resulting in the emergence of veritable “superbugs” which can threaten human health [5]. 

In this study, the detection of the *mcr-1* gene among *E. coli* « E1 » isolates susceptible to third-generation cephalosporins and carbapenems highlights the difficulty of surveillance of plasmid mediated resistance to colistin. In fact, colistin susceptibility testing is restricted to carbapenem-resistant isolates in our country suggesting the potential role of such isolates in the silent dissemination of the *mcr*-1 gene among other GNB.

The genetic environment of the *mcr-1* gene lacked the IS*Apl1* element, avoiding the threat of IS-mediated transmission. As previously described, IS*Apl1* was presumably involved in the initial transposition of the *mcr-1* element and then lost for stabilization of *mcr-1* on plasmids.

Colistin resistance was successfully transferred to the *E. coli* J53 strain via a conjugation experiment, suggesting that the *mcr-1* gene is located on a transferable plasmid. This result confirms the finding that the *mcr-1* gene is mobilized on plasmids that have spread to different *Enterobacteriaceae* [5]. 

## 5. Conclusions

In conclusion, colistin resistance was detected in 5% of the clinical Gram-negative bacteria collected from six Tunisian university hospitals. The main species found were *P. aeruginosa*, *E. coli* and *K. pneumoniae*. Colistin-resistant was associated with other resistance markers including ESBL (*bla*_CTX-M-15_) and or carbapenemases (*bla*_OXA-48_, *bla*_VIM_). We detected the emergence of sporadic *mcr*-1-positive *E. coli* strains in patients with urinary tract infections in Tunisia. Furthermore, the co-existence of colistin-resistant and ESBL genes; in this study suggests their possible spread to other human and environmental pathogens. Unfortunately, colistin susceptibility testing for non-carbapenem-resistant *Enterobacteriaceae* is not a routine procedure in diagnostic laboratories. Therefore, continuous colistin resistance surveillance studies are necessary, and the prescription of this antibiotic should be controlled. 

## Figures and Tables

**Table 1 antibiotics-11-01390-t001:** Distribution of GNB isolates according to their hospital origin.

Hospital’s Origin	Total Number of GNB ^a^ Isolates	CHROMagar™ COL-APSE Culture Positive	
		Number	Percentage
Bechir Hamza Children’s Hospital of Tunis (hospital 1)	234	46	20%
Charles Nicolle Hospital of Tunis (hospital 2)	213	24	11%
The Rabta Hospital (hospital 3)	180	5	3%
Habib Bourguiba University Hopsital of Sfax (hospital 6)	142	1	1%
Aziza Othmana Hospital of Tunis (hospital 4)	25	0	0%
National Bone Marrow Transplant Center (NBMTC) (hospital 5)	22	1	5%
Total	836	77	9.2%

^a^ GNB: Gram negative bacteria.

**Table 2 antibiotics-11-01390-t002:** Characteristics of colistin-resistant GNB isolates.

	*P. aeruginosa* (*n* = 28)	*E. coli* (*n* = 7)	*K. pneumoniae* (*n* = 5)	*Enterobacter cloacae* (*n* = 1)	*Raoutella terrigena* (*n* = 1)
Prevalence of colistin resistance, % (90% IC)	9.3 (6.5–12.1)	3 (1.2–4.8)	2.7 (0.8–4.7)	-	-
Colistin MIC ^a^ range, mg/L	4–128	4–32	32–128	32	32
*mcr*-1 gene detection, % (90% IC)	0	1.7(0.3–3.1)	0	0	0
ESBL ^b^, n (encoding genes)	0	3 (*bla*_CTX-M-15_)	5 (*bla*_CTX-M-15_)	0	1 (*bla*_CTX-M-15_)
Carbapenemase, n (encoding genes)	10 (*bla*_VIM_)	0	5 (*bla*_OXA-48_)	0	1 (*bla*_OXA-48_)
Clinical specimen, n (%)					
Urine	2	4	5	1	0
Blood	2	0	0	0	1
Wound	11	3	0	0	0
Sputum	5	0	0	0	0
Others ^c^	8	0	0	0	0
Total of isolates (*n* = 836) ^d^	301	233	183	25	5

^a^ MIC: Minimum Inhibitory Concentrations, ^b^ ESBL: extended-spectrum-beta-lactamase, Others ^c^: [Catheter (*n* = 2), Distal protected aspirate (*n* = 3), pleural fluid (*n* = 3)], Total of isolates (*n* = 836) ^d^: include 89 *Acinetobacter baumannii* colistin susceptible isolates.

**Table 3 antibiotics-11-01390-t003:** Characteristics of the four *E****.***
*coli* isolates carrying *mcr-1* and TCE1 *E. coli* transconjugant.

Strain	Date of Isolation	Date of Hospitalisation	Ward	Patient		Sample Source	Colistin MIC (µg/mL)	Other Resistance Profile	*bla* Genes	Replicon Types	Phylogenetic Groups
				Gender	Age (Year)						
**E1**	12 March 2019	08 March 2019	Urology	Female	74	Urine	16	AMP, TIC, NAL, CIP, TET	*bla* _TEM-1_	None	None
**TCE1**	NA	NA	NA	NA	NA	NA	4	AMP, TIC	*bla* _TEM-1_	ND	ND
**E2**	16 March 2019	15 March 2019	Urology consultation	Female	70	Urine	32	AMP, TIC, CTX, NAL, CIP, TET, CHL	*bla*_TEM-1_, *bla*_CTX-M-15_	FIB, Y	D
**E3**	15 March 2019	NA	Urology consultation	Male	72	Urine	16	AMP, TIC, CTX, NAL, CIP, TET, CHL	*bla*_TEM-__1_, *bla*_CTX-M-15_	FIB, Y	D
**E4**	17 March 2019	NA	Emergency	Male	43	Urine	32	AMP, TIC, CTX, NAL, CIP, TET, CHL	*bla*_TEM-1_, *bla*_CTX-M-15_	FIB, Y	D

AMP: ampicillin, CIP: ciprofloxacin, CHL: chloramphenicol, CTX: cefotaxim, NAL: nalidixic acid, TET: tetracyclin, TIC: ticarcillin, TC: Transconjugant, ND: not determined; NA: not applicable; MIC: Minimum Inhibitory Concentrations.

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
