# Peer review of "Tunisian Multicenter Study on the Prevalence of Colistin Resistance in Clinical Isolates of Gram Negative Bacilli: Emergence of Escherichia coli Harbouring the mcr-1 Gene"

_antibiotics, 2022, doi:10.3390/antibiotics11101390_

Round 1
Reviewer 1 Report
The manuscript by Ferjani et al analyzed prevalence of colistin resistance in Gram-negative bacteria in clinical isolates. From a total of 836 strains, 77 strains grew on CHROMagar plate and 42 were resistant to colistin. The 42 strains were further analyzed for the presence of mcr gene and other beta-lactamases. Conjugation assays were also carried out to test whether mcr genes are transferable. Overall the study addresses a important question of the prevalence of colistin resistance in Tunisia in Gram-negative bacteria which are not multi-drug resistant. However, for scientific rigor and reproducibility the methods section needs to elaborated and clearly state how the experiments were carried out. Following are some of my comments
1. How were the samples characterized initially? Please provide a detailed explanation of how the oxidase essay and API system assay were carried out.
2. Line 91: What is McF?
3. Line 101: what does ...... mean ?
4. Line 108: Need reference for CLSI guidelines
5. Provide a complete list of primers used in this study.
6. How was the genetic environment of mcr-1 gene evaluated? Provide more details.
7. Line 122: What does .... mean?
8. Line 137 - 139: How was the phylogenetic analysis carried out ? Provide a detailed explanation.
Author Response
Reviewer 1
Comments and Suggestions for Authors
The manuscript by Ferjani et al analyzed prevalence of colistin resistance in Gram-negative bacteria in clinical isolates. From a total of 836 strains, 77 strains grew on CHROMagar plate and 42 were resistant to colistin. The 42 strains were further analyzed for the presence of mcr gene and other beta-lactamases. Conjugation assays were also carried out to test whether mcr genes are transferable. Overall the study addresses a important question of the prevalence of colistin resistance in Tunisia in Gram-negative bacteria which are not multi-drug resistant. However, for scientific rigor and reproducibility the methods section needs to elaborated and clearly state how the experiments were carried out. Following are some of my comments
- How were the samples characterized initially? Please provide a detailed explanation of how the oxidase essay and API system assay were carried out.
Details on the oxidase essay and API system realisation were provided in the text:
Isolates were firstly identified based on colony morphology and Gram stain. Then all colonies were tested for oxidase by adding bacterial inoculum on the cotton tipped swab already impregnated by the N,N,N,N tetramethyl-paraphenylenediamine reagent. According to the results of Gram stain and oxidase test, the API 20E and API NE systems (bioMérieux, Marcy l’Etoile, France) were used for biochemical identification of Enterobacteriaceae and non-Enterobacteriaceae, respectively. API strips test containing 20 wells with dehydrated substrates to detect enzymatic activity. A bacterial suspension was used to rehydrate strip wells before incubation. All results test were compiled to obtain a profile number, which is then compared with profile numbers in the API-Web software to determine the identification of the bacterial species. E. coli ATCC 25922 and P. aeruginosa ATCC27853 were used as control strains.
- Line 91: What is McF?
It is McFarland.
- Line 101: what does ...... mean ?
MICs: Minimum Inhibitory Concentrations
- Line 108: Need reference for CLSI guidelines
We have included a reference for CLSI guidelines, as suggested.
- Provide a complete list of primers used in this study.
We have included a table (Table 1) of all primers used in our study.
- How was the genetic environment of mcr-1 gene evaluated? Provide more details.
We relied on previous studies by Veldman et al. and Maamar et al. to evaluate the flanking regions of the mcr-1 gene. Indeed, all isolated we examined with PCR using the same primers described ( table 1): to screen for the upstream presence of ISApl1, we have used primers ISApl1-mcr-F (5′ – TGGACATTGGGAAGCCGATA – 3′ ) and ISApl1-mcr-R (5′ –GCCA CAAGAACAAACGGACT – 3′ ) and to screen for the downstream presence of pap-2, we have used primers MCR1-END-F (5’-GTGCGAACATCAGTCCTTGA-3’) and PAP2-INT-R (5’-CGCAACCAGCAAGTAGATCA-3’).
- Line 122: What does .... mean?
We have screened genes encoded for beta-lactams resistant determinant. Indeed, for strains producing extended spectrum beta-lactamases were screened for the presence of blaCTX-M, blaTEM, and blaSHV genes.
- Line 137 - 139: How was the phylogenetic analysis carried out ? Provide a detailed explanation.
Phylogenetic group was determined by PCR using primers listed in table 1 as described by Clermont et al. 2013.
Reviewer 2 Report
In this manuscript, the authors are investigating the epidemiology of plasmid mediated colistin resistance in Gram negative bacteria isolated from clinical setting. Authors report emergence of mcr-1 carrying clinical E. coli strains responsible for UTIs in Tunisia. Thus, they propose more frequent surveillance for colistin resistance irrespective of the resistance to beta lactams and or carbapenem.
Following are my detailed comments
Line 28: Extra space after "spread"
Line 30: Fix spacing "5.02%"
Line 33: Unnecessary dots after infection
Line 61: Extra space after "is"
Line 65: harbouring
Line 66-67: Needs to be rephrased. Make it clear that unawareness is of colistin resistance
Line 87: Colony morphology
Line 101, 122, 159: Unnecessary punctuation of multiple dots/periods
Line 168: responsible for
Line 177-180: No data shown for transconjugation. A supporting figure of PCR/ electrophoresis corroborating successful transconjugation would strengthen the findings. More background information is needed before introducing data with respect to pap2 and ISAp11
Line 199: Sentence needs to be rephrased
Line 201: still "remains" underestimated
Line 219-221: Describe more on how the authors are reaching to the conclusion about dissemination outside the hospitals
Line 226: What does " genetics action taken" suggest? Does author mean genetic alterations?
Line 232: Pasmid mediated instead of plasmidic
Line 235: space between " avoiding the"
Author Response
Reviewer 2
In this manuscript, the authors are investigating the epidemiology of plasmid mediated colistin resistance in Gram negative bacteria isolated from clinical setting. Authors report emergence of mcr-1 carrying clinical E. coli strains responsible for UTIs in Tunisia. Thus, they propose more frequent surveillance for colistin resistance irrespective of the resistance to beta lactams and or carbapenem.
Following are my detailed comments
Line 28: Extra space after "spread"
Done
Line 30: Fix spacing "5.02%"
Done
Line 33: Unnecessary dots after infection
Done
Line 61: Extra space after "is"
Done
Line 65: harbouring
We have corrected.
Line 66-67: Needs to be rephrased. Make it clear that unawareness is of colistin resistance
Line 87: Colony morphology
We have changed as recommended.
Line 101, 122, 159: Unnecessary punctuation of multiple dots/periods
Done
Line 168: responsible for
We have included the modification, as suggested.
Line 177-180: No data shown for transconjugation. A supporting figure of PCR/ electrophoresis corroborating successful transconjugation would strengthen the findings. More background information is needed before introducing data with respect to pap2 and ISAp11
We have included the gel electrophoresis figure (Figure 1) of the mcr-1 PCR products of donor and transconjugant strains, as suggested.
Line 199: Sentence needs to be rephrased
We have rephrased the sentence, as recommended.
Line 201: still "remains" underestimated
We have included the modification, as suggested.
Line 219-221: Describe more on how the authors are reaching to the conclusion about dissemination outside the hospitals
The following sentence was added
According to the medical record review about patients acquired E. coli harbouring the mcr-1 gene, the onset of patient symptoms has occurred before the consultation of emergency and urology wards. These data suggested the community origin of the urinary tract infection caused by isolates harbouring the mcr-1 gene. In order to support this hypothesis concluded during this short surveillance period, further studies are needed to evaluate the spread of plasmid colistin resistance outside hospitals and control their transmission.
Line 226: What does " genetics action taken" suggest? Does author mean genetic alterations?
The following sentence was added:
Integration of mcr-1 into the bacterial chromosome occurs in some strains.
Line 232: Pasmid mediated instead of plasmidic
We have included the modification, as suggested.
Line 235: space between " avoiding the"
Done